# Study of the Pattern Preparation and Performance of the Resistance Grid of Thin-Film Strain Sensors

**DOI:** 10.3390/mi13060892

**Published:** 2022-06-01

**Authors:** Yunping Cheng, Wenge Wu, Yongjuan Zhao, Yanwen Han, Ding Song

**Affiliations:** School of Mechanical Engineering, North University of China, Taiyuan 030051, China; ypchengbk@163.com (Y.C.); zyj@nuc.edu.cn (Y.Z.); hyw9226@126.com (Y.H.); sd199402@163.com (D.S.)

**Keywords:** thin-film strain sensor, Ni80Cr20 alloy film, pattern preparation, process parameters, orthogonal test

## Abstract

The thin-film strain sensor is a cutting-force sensor that can be integrated with cutting tools. The quality of the alloy film strain layer resistance grid plays an important role in the performance of the sensor. In this paper, the two film patterning processes of photolithography magnetron sputtering and photolithography ion beam etching are compared, and the effects of the geometric size of the thin-film resistance grid on the resistance value and resistance strain coefficient of the thin film are compared and analyzed. Through orthogonal experiments of incident angle, argon flow rate, and substrate negative bias in the ion beam etching process parameters, the effects of the process parameters on photoresist stripping quality, etching rate, surface roughness, and resistivity are discussed. The effects of process parameters on etching rate, surface roughness, and resistivity are analyzed by the range method. The effect of substrate temperature on the preparation of Ni Cr alloy films is observed by scanning electron microscope. The surface morphology of the films before and after ion beam etching is observed by atomic force microscope. The influence of the lithography process on the surface quality of the film is discussed, and the etching process parameters are optimized.

## 1. Introduction

As the development of the micro-machine field rapidly increases, the requirements for machining accuracy in machine manufacturing are also increasing. Cutting force, which is an important parameter in the metal cutting process, directly affects both the work-piece quality and the tool life. As a result, it is particularly important to accurately measure the cutting force. In recent years, researchers have conducted a significant amount of work on cutting force [1,2,3,4]. The different types of cutting dynamometers mainly include strain dynamometers, piezoelectric dynamometers, current dynamometers, and capacitive dynamometers, etc. Among these, strain dynamometers and piezoelectric dynamometers are widely used. The piezoelectric dynamometer measures the cutting force through the piezoelectric effect of piezoelectric crystals [5,6]; it has the advantages of high rigidity and good dynamic characteristics, but it is sensitive to strong magnetic field interference, and charge leakage will occur when the humidity is high. The strain dynamometer measures the cutting force through the strain effect of film resistance [7], which has the characteristics of high rigidity, excellent dynamic characteristics, strong anti-interference ability, and good linearity.

Due to its small size and high precision, the film strain sensor can be used in embedded development and has become one of the main directions of sensor development [8,9,10,11,12]. Alloy film strain sensors are generally divided into the following, according to their functions: substrate, insulating layer, sensitive layer, electrode area, and protective layer. The thin-film sensitive layer is the core part of the thin-film sensor. The material properties, the shape and size of the sensitive layer, and its electrical and mechanical properties are the key to determining the pros and cons of the thin-film sensor. Ni80Cr20 film is used for thin-film resistors and sensing materials of commonly used thin-film strain gauges due to its high reliability, high resistivity, high sensitivity coefficient, and low temperature coefficient of resistance (TCR) [13]. Ni80Cr20 alloy film needs to be processed in the form of a resistance grid as the strain sensitive layer, so it is necessary to pattern the film.

The film patterning method mainly includes the following: Chijui Han et al. used micro-molding and stamping to manufacture thin-film resistor grids [14]. This method improves the preparation efficiency, but the stamping process changes the grain arrangement of the surrounding film, increases the internal stress of the film, and thereby reduces the surface quality of the film. Shuwen Jiang et al. used a metal reticle method to pattern the resistive layer of the thin-film sensitive layer [15], but there is a certain limit on the width of the resistive gate. The metal mask is difficult to fabricate and is not suitable for the preparation of the fine film resistor grid.

At present, the resistance grid is mainly obtained by lithography and ion beam etching. The film patterning method mainly includes the following: First, the most common method is to form an anti-pattern photoresist pattern by photolithography, then deposit a sensitive layer through a deposition system and remove the photoresist to form a desired pattern [16]; this method of preparing a resistive gate is the same as photolithography magnetron sputtering (PMS). IBE is a well-established technique for patterning magnetic and precious metals in memory applications, and it has the characteristics of non-selectivity, accuracy, directivity, high resolution, and flexible processing [17,18]. Narasimhan Srinivasan et al. studied the relationship between ion beam etching uniformity, angle related etching rate, and etching surface quality, including the surface quality and accuracy of thin-film grids [19]. Abdullaev et al. studied ion beam etching of dense porous PZT films under different Ar+ ion flux incidence angles. The effects of etching rate and incidence angle on the microstructure of the films were discussed [20]. Soyer et al. selected the film material and discussed the relationship between the etching rate of PZT on silicon substrate and process parameters (gas composition, current density, and accelerating voltage), though some electrical damage was produced in the etching process. The evolution of electrical properties has been studied when a metal mask protects the PZT layer [21,22]. Gallium oxide thin films were deposited on sapphire substrates at different temperatures by laser pulse laser deposition. It was found that, with the increase in substrate temperature, the crystallinity of the thin film increased and the etching rate decreased [23]. In the preparation scheme of the film strain sensor sensitive layer resistance grid, the use of photolithography and IBE of the film patterning method can improve the process of magnetron sputtering Ni Cr film, thereby improving the film properties. This method of preparing a resistive gate by photolithography and ion beam etching is the same as photolithography ion beam etching (PIBE).

In this paper, the two film patterning methods of photolithography magnetron sputtering and photolithography ion beam etching are compared and analyzed, and the influence of the structure size of the thin-film resistance grid prepared by the two methods on the film performance is discussed.

## 2. Experimental Design

### 2.1. Thin-Film Resistance Grid Patterning Method Comparison Test Design

In this experiment, IBE is applied as the preparation method of the alloy film resistance strain sensor sensitive grid and compared with the previous resistance grid preparation method of PMS. The preparation process of the two sets of thin-film resistor grids is shown in Figure 1 and Figure 2. The experimentally prepared alloy film strain sensor is mainly used for tensile experimental research, and the shape of the substrate is the same. First, a surface treatment was performed on a stainless-steel substrate with a thickness of 0.55 mm, and the preparation of the insulating layer was completed by sputtering a thickness of 600 nm of Al_2_O_3_ and a thickness of 300 nm of Si_3_N_4_ onto the surface [24]. Figure 1 shows the film patterning process of PMS. On the stainless-steel substrate with an insulating layer on the surface, photolithography is used to develop and remove the photoresist pattern on the insulating layer substrate, and the thickness of the adhesive layer is approximately 2.5 μm. A layer of Ni Cr film with a thickness of 800 nm was then prepared by magnetron sputtering, and the photoresist and the film on the glue were then peeled off to complete the fabrication of the resistance layer of the sensitive layer.

In the process of preparing the thin-film resistance grid of the sensitive layer of the thin-film sensor by PMS, the photoresist with the substrate will fail at 150 °C and above, so the preparation process of the subsequent sensitive layer film is highly required.

Figure 2 shows the film patterning process of PIBE. First, a layer of Ni Cr film is sputtered onto a stainless-steel substrate with an insulating layer on the surface by magnetron sputtering. The photoresist is then developed on the Ni Cr film by photolithography to form a resistance grid-shaped pattern. Next, a thin-film resistor grid is etched by IBE technology. Finally, the remaining photoresist is stripped with a cleaning agent, such as acetone, to complete the fabrication of the resistor grid. An example of a film mask used for lithography exposure in the two processes is shown in Figure 3.

### 2.2. Resistance Value and Resistance Strain Coefficient with Different Geometric Dimensions of Thin-Film Strain Sensors Prepared by PMS and PIBE

(1) The length of the longitudinal grid of the thin-film strain sensor

A semiconductor characterization system is used to measure the resistance of thin-film strain sensors with different lengths of longitudinal grid obtained by PMS and PIBE. The measurement results are averaged by three measurements. The results are shown in Table 1. The error and range of the measured values are analyzed.

From the resistance measurement results in Table 1, it can be seen that the resistance values of the thin-film strain sensors by the two methods have stability. The resistance value range of 3–5 mm length in PMS is stable within 20.4, and the resistance value range in PIBE is 26.1. It can be observed that the average resistance error multiple increases slightly from 3.5 mm to 4.1 mm in PMS, and that increases from 2.1 to 2.4 in PIBE. In the two methods, the thin-film resistance grid sample with a resistance grid length of 3 mm offers the best result on the total resistance, single grid range, and average error multiple, and the average error multiples are 3.523 and 2.165, respectively.

The resistance strain coefficient of the thin-film sensor needs to be measured and analyzed by tensile strain. A digital display push–pull meter is used for the tensile test, and a signal acquisition instrument (DASP) is used for data acquisition and analysis. The thin-film sensor is connected to the signal acquisition system to form a 1/4 Wheatstone bridge circuit. The wiring of the thin-film sensor with elastic substrate is shown in Figure 4.

According to the tensile strain, the theoretical tensile strain of the elastic substrate of the thin-film sensor is calculated by Equation (1).
(1)$$\varepsilon =\frac{\Delta l}{l}=\frac{F}{EA}$$
where *E* = 195 Gpa is the elastic modulus and *A* is the cross-sectional area of the measured position.

Within the range of elastic deformation, when the safety factor of 304 stainless steel is 2, the allowable stress [σ] = 102.5 MPa and the elastic base can withstand the maximum tensile force *F*_max_ = 451 N. Therefore, the range of 0–400 n is selected for tensile test.

The signal acquisition instrument (DASP) can match the voltage with the resistance value and maximize the output response. The equipment can directly output the voltage change and strain value. The resistance change can be obtained through the voltage change. The correlation is shown in Formula (2), in which *U_i_* is the input voltage and *U_0_* is the output voltage.
(2)$$U_{i}=\frac{1}{4}\frac{\Delta R}{R}U_{0}$$

The resistance variation and resistance stability of thin-film strain sensors with different lengths of longitudinal grid under tensile test are analyzed. The tensile results are shown in Figure 5.

As shown in Figure 5, linear fitting and strain coefficient calibration are carried out on the measurement results of the thin-film strain sensor by different methods. The slope of the fitting curve is the resistance strain coefficient of the thin-film strain sensor. The resistance strain coefficient decreases gradually with the increase in the length of the longitudinal grid when it is in the range of 3–6 mm. The resistance strain coefficient of the thin-film resistance grid is between 1.46–1.62 and 1.54–1.62, respectively, which is within the range of the Ni-Cr material strain coefficient. Among them, the maximum strain coefficients of the strain grid with a longitudinal grid length of 3 mm are 1.61 and 1.62, respectively.

The resistance values of thin-film strain sensors by the two methods have stability. Among them, the resistance error of the thin-film resistance grid prepared by PIBE is significantly lower than that of PMS; the difference between the errors of two resistance multiples is almost 1.5. Because the adjustment of substrate temperature is increased in the process of preparing Ni-Cr thin-film by PIBE, the resistance value of thin-film resistance grid is optimized to a certain extent.

According to Figure 5, the resistance strain coefficients of thin-film strain sensors prepared by PIBE are higher than those prepared by PMS. Resistance value and the geometric size of the thin-film resistance grid are the two main aspects affecting the resistance strain coefficient; the influence of the length of the thin-film resistance grid on the strain coefficient is very small compared with that of thin-film resistance, and the main reason for the increase in strain coefficient is the lower resistance value of the thin-film strain sensor.

(2) The width of the longitudinal grid of the thin-film strain sensor

The thin-film strain sensors have longitudinal grids with widths of 0.05 mm, 0.1 mm, 0.15 mm, and 0.2 mm. The resistance value of the thin-film strain sensor is 1200 Ω, and the other dimensions of the thin-film resistance grids are shown in Table 2.

The semiconductor characterization system (4200 SCS) is used to measure the resistance of thin-film strain sensors with longitudinal grids of different widths. The measurement results, range, and error analysis results are shown in Table 3.

As can be seen from the resistance measurement results in Table 2, the resistance values of thin-film strain sensors by the two methods have stability. The resistance value range of 0.05–0.2 mm width in PMS is stable within 21.5, and the resistance value range in PIBE is 33.1. The other three single grid ranges are less than 21.1 in PIBE. With the increase of resistance grid width from 0.05 mm to 0.2 mm, the total resistance and the average error multiple gradually decrease. For thin-film resistance grids with widths of 0.05 mm to 0.2 mm, the average error multiples are 3–5.2 and 2.8–4, respectively. The thin-film resistance grid sample of 0.2 mm wide performs the best on the total resistance, single grid range, and average error multiple, with an average error multiple of 3.045. The thin-film resistance grid sample with the 0.1 mm width performs the best on the total resistance, single grid range, and average error multiple, and the average error multiple is 2.139.

As shown in Figure 6, linear fitting and strain coefficient calibration were carried out on the measurement results of the thin-film strain sensor by different methods. With the increase in width, the resistance strain coefficients of the two methods first increase and then decrease when the width of the grid is in the range of 0.05–0.2 mm. The resistance strain coefficients of thin-film strain sensors are between 1.45–1.55 and 1.45–1.62, respectively, which are in the range of Ni-Cr material strain coefficients. Among them, the maximum strain coefficients of the strain grid with a longitudinal grid width of 0.1 mm are 1.55 and 1.62, respectively.

According to Figure 6, the resistance with different widths of thin-film strain sensors prepared by the two methods is roughly similar to the results of different lengths. With the same width, the resistance error of the thin-film strain sensors prepared by PIBE is lower than that of PMS, and the difference of resistance error multiple between the two is almost 1.5. With the increase in the width, the resistance value of the thin-film sensor prepared by PIBE first decreases and then increases. The change of the width of the thin-film sensor has a significant effect on the resistance strain coefficient of the thin-film sensor. The resistance strain coefficient of the thin-film strain sensor prepared by PIBE is generally higher than that by PMS.

(3) The thickness of the longitudinal grid of the thin-film strain sensor

The thin-film strain sensors have longitudinal grids with thicknesses of 800 nm, 900 nm, 1000 nm, and 1100 nm. The resistance value of the thin-film strain sensors is 1200 Ω, and the other dimensions of the thin-film resistance grids are shown in Table 4.

From the resistance measurement results in Table 5, it can be seen that the resistance values of the thin-film strain sensors by the two methods have stability. With the resistance grid thickness increase from 800 nm to 1100 nm, the total resistance value gradually decreases. The minimum range of the 1000-nm thick thin-film resistance grid is 14 in PMS, and the minimum range of the 900-nm thick thin-film resistance grid is 14.1 in PIBE. With the increase of resistance grid thickness from 800 nm to 1100 nm, the average error multiple decreases and then increases, and the values are between 3.6–4.4 and 2.1–2.5, respectively. When the thickness of the thin-film resistance grid sample is 1000 nm, the total resistance, single grid range, and average error multiple are better, and the average error multiples are 3.632 and 2.125, respectively.

The resistance variation and resistance stability of thin-film strain sensors with different thicknesses of grid under tensile test are analyzed. The tensile results are shown in Figure 7.

As shown in Figure 7, linear fitting and strain coefficient calibration were carried out on the measurement results of the thin-film strain sensor by different methods. With the increase in thickness, the resistance strain coefficient gradually increases, and the resistance strain coefficient of the thin-film resistance grid is between 1.45–1.56 and 1.57–1.63, respectively, which are in the range of the strain coefficient of Ni-Cr material. Among them, the maximum strain coefficients of the strain grating with the thickness of 1100 nm are 1.56 and 1.63, respectively.

Among them, the resistance error of the thin-film resistance grid prepared by photolithography is still significantly lower than that prepared by photolithography magnetron sputtering, and the difference between the two resistance error multiples is about 1.5. With the increase in thickness, the resistance of the thin-film resistance grid decreases gradually. The change of the thickness of the thin-film sensor has a significant effect on the resistance strain coefficient of the thin-film sensor. The resistance strain coefficient of the thin-film strain sensor prepared by PIBE is generally higher than that by PMS.

### 2.3. Orthogonal Experiment on Process Parameters of PIBE

In the pattering process of the Ni Cr film of PIBE, the cathode current is 5.8 A and the neutralization current is 6.10 A. Both are the matching current generated according to the change of voltage, and the regulation effect is limited; the ion beam energy is the accumulation of the arc voltage and the negative bias of the substrate. The arc voltage is 45 V, which has little effect. In the process, the negative bias of the substrate can be adjusted to change the properties of the film. The acceleration voltage is 300 V, and the coupling coefficient is 1.25.

As a result, the incident angle of the ion beam, argon flow rate, and substrate negative bias are the three main process parameters of ion beam etching. Table 6 is the factor level table of the orthogonal test of the three process parameters.

Table 7 shows the factor level orthogonal test table L9 (33) in Table 2 and the test results. The etching rates of Ni Cr alloy and photoresist are obtained by the thickness dividing time, and the roughness is measured by optical microscope.

In the process of ion beam etching, AZ6140 photoresist is used as the mask material to form the resistance grid pattern of the Ni-Cr film. It can be seen from Table 7 that, under the same process parameters, the etching rate of photoresist is 1 nm/min–3.7 nm/min lower than that of Ni-Cr film. The etching rate of photoresist increases with the increase in the negative bias of the substrate, and the increase value of the etching rate of photoresist changes in the range of 2 nm when the negative bias of substrate increases by 100 V. At the same time, the ratio of the etching rate of photoresist to the ion beam etching of Ni-Cr film is between 0.86 and 0.96; as a result, AZ6140 photoresist can meet the conditions of ion beam etching.

The etched photoresist was subjected to a peeling test. When the negative bias voltage of the substrate was 550 V, the etched photoresist had incomplete peeling, such as in Test Nos. 3, 5, and 7. This can be explained by the fact that when the negative bias voltage of the substrate is high, the ion beam energy would be increased, resulting in the carbonization of the photoresist. As a result, the negative bias voltage of the base should be less than 550 V in order to reduce the adverse impact of photoresist peeling on the nickel chromium film.

## 3. Results and Analysis

### 3.1. Resistance Strain Coefficient and Error Analysis of Resistance Grid

Statistical analysis was carried out according to the four levels of each size parameter design. Figure 8 shows the influence of the geometric size of the grid on the resistance value of the thin-film resistance grid by the two preparation methods. As show in Figure 8a,b, the width of the longitudinal grid has the highest impact on the resistance value of the thin-film resistance grid, and the resistance error generally decreases with the increase in the width; with the increase in the thickness, the resistance error first decreases and then increases, and it reaches the lowest when the thickness is 1000 nm. Relating to the length of the thin-film resistance grid, in Figure 8b, the curve of error fluctuates slightly multiple times up and down, and this shows a stable trend. The error multiple of the length of the resistance grid prepared by the two methods is relatively minimal.

The resistance strain coefficients of the thin-film resistance grid prepared by the two preparation methods are shown in Figure 9.

As Figure 9 shows, the resistance strain coefficient of the width of the longitudinal grid obtained by the two methods first increases and then decreases, and the corresponding resistance strain coefficient value is the largest when the width is 0.1 mm. The resistance strain coefficient increases as the thickness of the grid prepared by PMS increases, and the resistance strain coefficient first decreases and then increases as the thickness of the grid obtained by PIBE increases. The resistance strain coefficient first decreases and then increases when the length of the resistance grid prepared by PMS increases, and the resistance strain coefficient decreases when the thickness of the resistance grid obtained by PIBE increases. The thickness of the longitudinal grid has the least effect on the resistance strain coefficient. According to the above analysis, it can be seen that a better resistance strain coefficient can be obtained by PIBE. For the resistance grid, the cutting force is mainly measured in the length direction of the grids; the PIBE is used to obtain the resistance grid in this paper.

### 3.2. Comparative Analysis of Thin-Film Resistor Grid Preparation Methods

The thin-film resistor grids prepared by PIBE and the thin-film resistor grids prepared by PMS are analyzed for electrical properties and surface morphology. Thin-film resistor grids with a resistance of 1200 Ω are selected. By preparing thin-film resistor grids with different widths, the influence of the width of the resistor grid on the film resistance is analyzed, and the film size is determined by Formula (3).
(3)$$R=\rho \frac{L}{d\times h}$$

Some of the thin-film resistor grids produced by the PIBE are shown in Figure 10a. The semiconductor characterization system 4200 SCS is used to measure the resistance of the thin-film resistor gate. The experimental equipment is shown in Figure 10b.

Three samples are selected and measured three times to obtain the average value. The final measurement results and errors are shown in Table 8. It can be seen from Table 8 that, under the same width, the average error multiple of the resistance value of the resistance gate prepared by the PIBE method is lower than that of the PMS method. Through the overall longitudinal observation, the accuracy of the resistance value of the thin-film resistance gate increases as the width of the resistance gate increases.

SEM is used to observe the size, surface morphology, and contour integrity of the thin-film resistor grid prepared by the PIBE method, as shown in Figure 11. Figure 11a shows a thin-film resistor grid sample of 0.1 mm width prepared by the PIBE method, and Figure 11b shows a thin-film resistor grid sample of 0.15 mm width prepared by the PIBE method. The overall structure is complete, with very few missing and redundant areas. In addition, the grid size is uniform and stable.

### 3.3. Effects of Etching Process Parameters on Etching Rate, Surface Quality, and Resistivity

According to the orthogonal test results in Table 7, the effects of various factors on the film etching rate, film surface roughness, and resistivity are discussed by range analysis, as shown in Table 9, where X is the incident angle, Y is the argon flow, Z is the negative bias voltage of the substrate, ki (i = 1, 2, 3) is the mean value of the measurement results at every various levels, and R_j_ (j = A, B, C) is the range of various factors. From the results in the Table 9, the ki of X, Y, and Z have small changes and change in the range of 24.5–29.4, but the R_j_ of X, Y, and Z have great changes and change in the range of 0.9–9.1. As a result, the negative bias voltage has the greatest impact to the film etching rate. Similarly, the incident angle has the greatest impact on the film roughness, and the incident angle has the greatest impact on the film resistivity.

Figure 12 is a parameter range analysis of the film etching rate, roughness, and film resistivity; they are the results of process parameters such as incident angle, argon flow, and substrate negative bias.

Figure 12a shows the range analysis of thin-film etching rate. The influence order of process parameters on the etching rate is s follows: substrate negative bias > argon flow > incident angle. Because the substrate negative bias can be superimposed with arc electrode voltage to form ion source energy, increased ion beam energy accelerates the etching speed. The increase in argon flow increases the number of argon ions and accelerates the etching speed similarly. Figure 12b shows the range analysis of film surface roughness. The influence order of process parameters is as follows: incident angle > substrate negative bias > argon flow. Changing the incident angle of the ion beam can effectively improve the surface quality of the etching surface. When the incident angle is 45°, the surface roughness value is the smallest and the surface quality is the best. Figure 12c shows the range analysis of resistivity. The influence order of process parameters is as follows: incidence angle > substrate negative bias > argon flow. The resistivity is the smallest when the incidence angle is 45° and the resistivity is the largest when the incidence angle is 20°, indicating that the incidence angle would change the distance between thin-film particles and cause the change of resistivity.

### 3.4. Variance Analysis of Thin-Film Etching Process Parameters

The regression equations of etching rat, e V, surface roughness, Ra, and resistivity on incident angle, substrate negative bias, and argon flow are established using Minitab software and the multiple regression fitting method, as shown in Formula (4). The analysis of variance of the regression equation is shown in Table 5.
(4)$$\{\begin{array}{c} v=-1.14-0.0160X+0.833Y+0.0455Z\\ Ra=23.6-0.0600X+0.220Y+0.0267Z\\ \rho =0.545-0.0034X+0.0015Z+0.032Z \end{array}$$

According to Table 10, when the inspection level is taken as α = 0.005, the F distribution table is checked to obtain F_α_ (3,5) = 16.53. Compared with the data of F in the table, the response regression models of etching rate, V, have reached a significant level, but there is a large gap between the response regression models of surface roughness and resistivity.

In order to obtain a good surface quality of the Ni-Cr alloy film and stable film deposition rate, the etching process parameters are as follows: 45° ion beam incidence angle, 7.5 sccm argon flow and 450 V substrate negative bias, 25.3 nm/min etching rate of Ni-Cr alloy thin-film, and 23.4 nm/min etching rate of AZ6140 photoresist.

### 3.5. Effect of Substrate Temperature on Ni Cr Alloy Film

When the resistance grid is prepared by the PMS method, the photoresist is coated before the preparation of the Ni Cr film, and the photoresist fails at 150 °C. When the resistance grid is prepared by PIBE, the photoresist is coated after the preparation of the nickel chromium film, which will not be affected by temperature. Figure 13 shows the surface morphology of Ni Cr film without and with substrate temperature by magnetron sputtering by scanning electron microscopy (SEM). Figure 13a is an SEM of the Ni Cr film without adding substrate temperature. The surface of the Ni Cr film has an obvious grain size, and the overall particle growth is uneven, resulting in low surface flatness, dark gloss, and large surface roughness. Figure 13b is an SEM of the Ni Cr film, adding the substrate temperature, which is higher than 200 °C. The grain growth of the Ni Cr film is relatively dense, and the gloss is bright, so the overall surface is even and flat.

Figure 14 shows the surface topography of the Ni Cr film by atomic force microscopy (AFM). As shown in Figure 14a, the surface of the Ni Cr film has more and higher peaks, so the roughness is higher. However, the surface of the Ni Cr film in Figure 14b has fewer and lower peaks, and the roughness is lower. Figure 14c shows that the film surface is relatively uniform as a whole, with small and dense wave peaks. As a result, substrate temperature and ion beam etching can improve the surface quality of Ni Cr alloy films.

## 4. Conclusions

In this paper, photolithography and ion beam etching are combined to pattern Ni Cr alloy film in a thin-film strain sensor to obtain the resistance grid structure, which is a new method of obtaining high-quality patterns on the metal film of thin-film strain sensors.

(1) PIBE is more suitable for Ni Cr alloy film patterning than PMS, because substrate temperature during the preparation of the Ni Cr film in PMS would fail at high temperatures. By comparing the single gate range, the average error multiples and resistance strain coefficients of the Ni-Cr alloy resistance grid obtained by PIBE are better than those obtained by PMS. Additionally, the stability of the resistance value of the thin-film resistor gate prepared by the PIBE method is better than that of the PMS method.

(2) In the PIBE, the influence order of the etching process parameters on the etching rate is as follows: substrate negative bias > argon flow > incident angle. The influence order on the film surface roughness is as follows: incident angle > substrate negative bias > argon flow. The influence order on the resistivity is as follows: incident angle > substrate negative bias > argon flow. The resistivity is smallest when the incident angle is 45°, and resistivity is the largest when the incident angle is 20°. Through the analysis of variance and considering the influence of stripping photoresist, the etching process parameters can be 45° ion beam incidence angle, 7.5 sccm argon flow, and 450 V substrate negative bias; the etching rate of nickel chromium alloy film is 25.3 nm/min, and the etching rate of AZ6140 photoresist is 23.4 nm/min.

(3) The test results of SEM and AFM show that increasing the substrate temperature can refine the surface grains, and PIBE can make the film surface of the Ni Cr alloy resistance grid smooth, the film surface is relatively uniform, and the surface roughness value is small.

## Figures and Tables

**Figure 1 micromachines-13-00892-f001:**
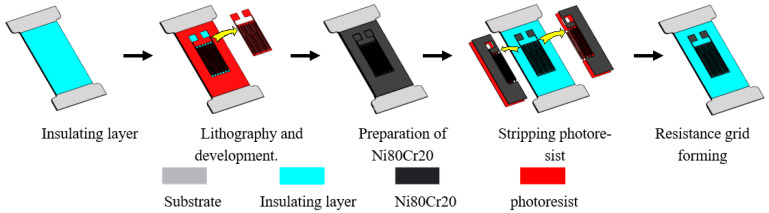
The film pattering process of PMS.

**Figure 2 micromachines-13-00892-f002:**
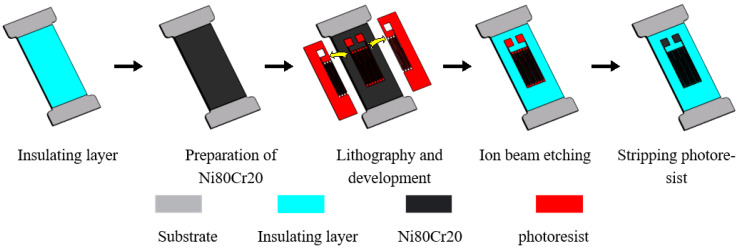
The film pattering process of PIBE.

**Figure 3 micromachines-13-00892-f003:**
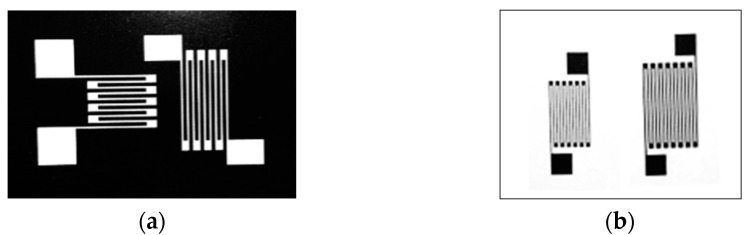
Photolithography process mask. (**a**) Film paper used in PMS. (**b**) Film paper used in PIBE.

**Figure 4 micromachines-13-00892-f004:**
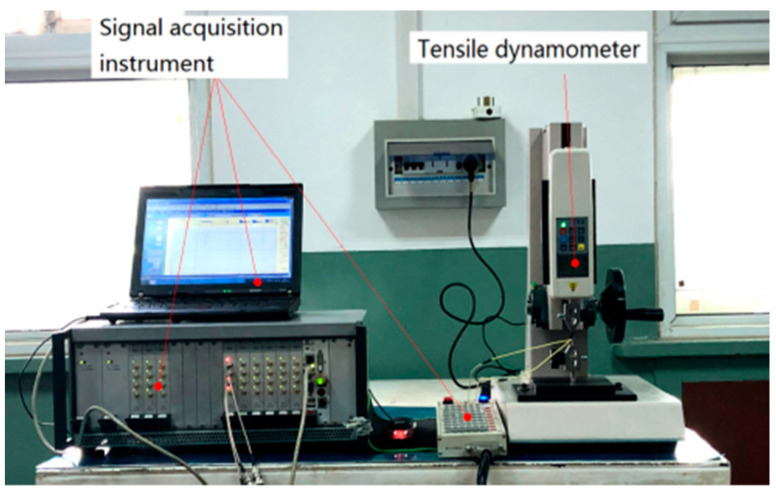
Thin-film resistance grid tensile test diagram.

**Figure 5 micromachines-13-00892-f005:**
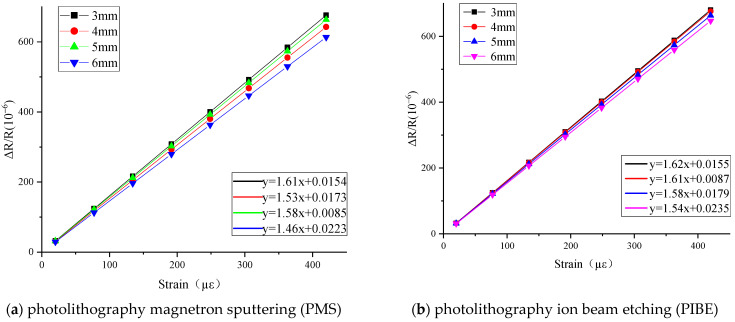
Tensile results of different grid lengths prepared by PMS and PIBE.

**Figure 6 micromachines-13-00892-f006:**
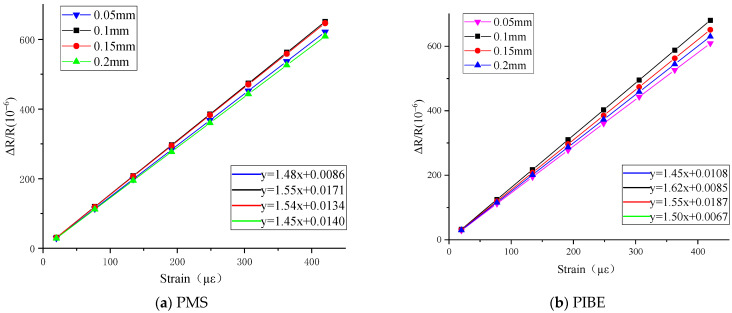
Results of thin-film strain sensors with different grid widths prepared by PMS and PIBE.

**Figure 7 micromachines-13-00892-f007:**
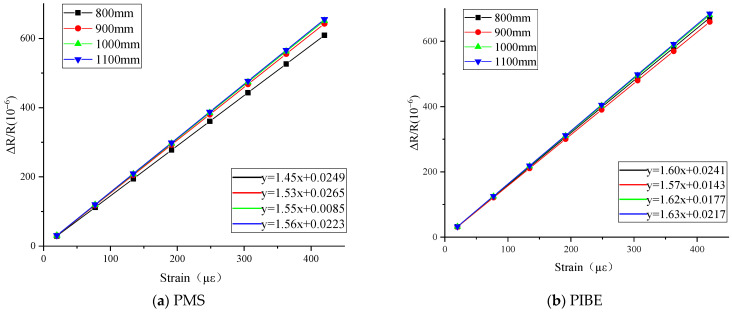
Results of thin-film strain sensors with different grid thicknesses prepared by PMS and PIBE.

**Figure 8 micromachines-13-00892-f008:**
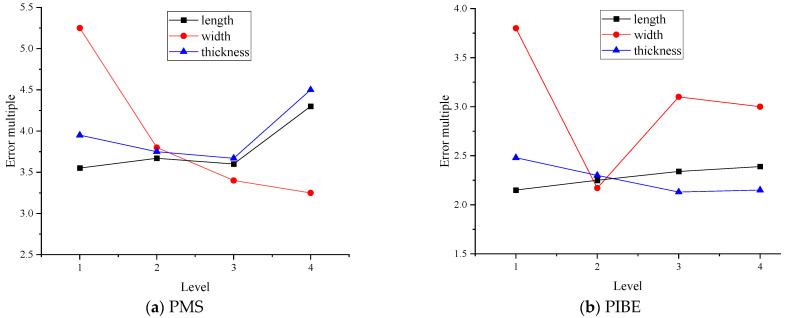
Analysis of resistance values of thin-film resistor grid geometry.

**Figure 9 micromachines-13-00892-f009:**
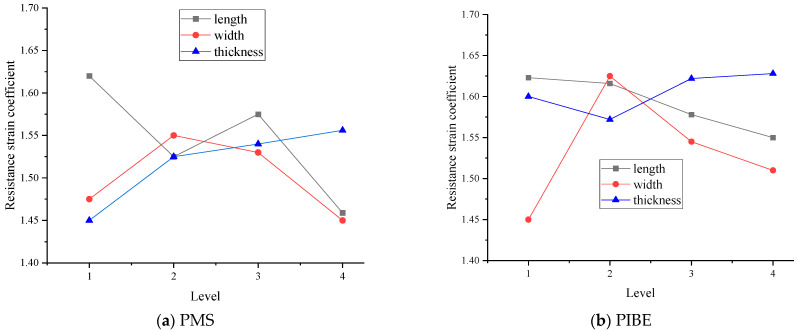
Analysis of resistance strain coefficients of thin-film resistor grid geometry.

**Figure 10 micromachines-13-00892-f010:**
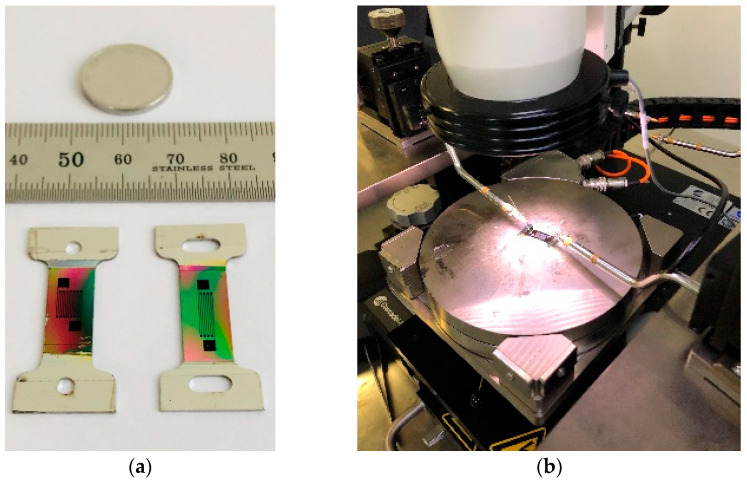
Film resistance gate resistance measurement experiment. (**a**) Physical map of thin-film resistor grid. (**b**) Thin-film resistor grid resistance measurement.

**Figure 11 micromachines-13-00892-f011:**
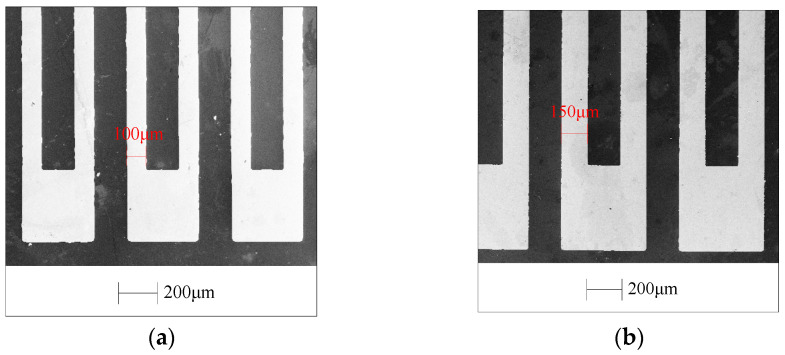
SEM image of thin-film resistor grid sample. (**a**) SEM image of 0.1 mm thin-film resistor grid. (**b**) SEM image of 0.15 mm thin-film resistor grid.

**Figure 12 micromachines-13-00892-f012:**
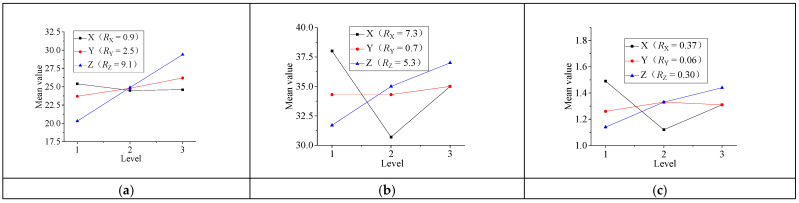
Parameter range analysis of film etching rate, roughness, and film resistivity. (**a**) Parameter range analysis of film etching rate. (**b**) Parameter range analysis of film roughness. (**c**) Parameter range analysis of film resistivity.

**Figure 13 micromachines-13-00892-f013:**
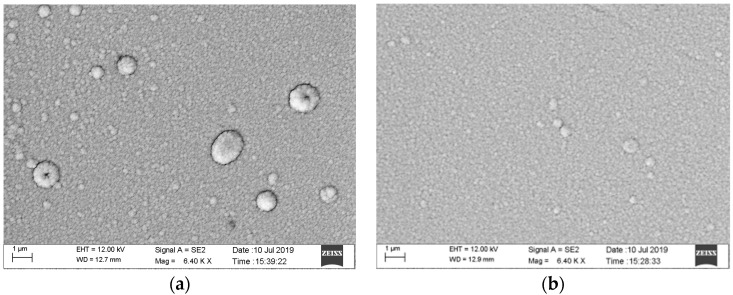
SEM image of Ni Cr thin-film. (**a**) SEM image of NiCr thin-film without surface temperature. (**b**) SEM image of NiCr thin-film with surface temperature.

**Figure 14 micromachines-13-00892-f014:**
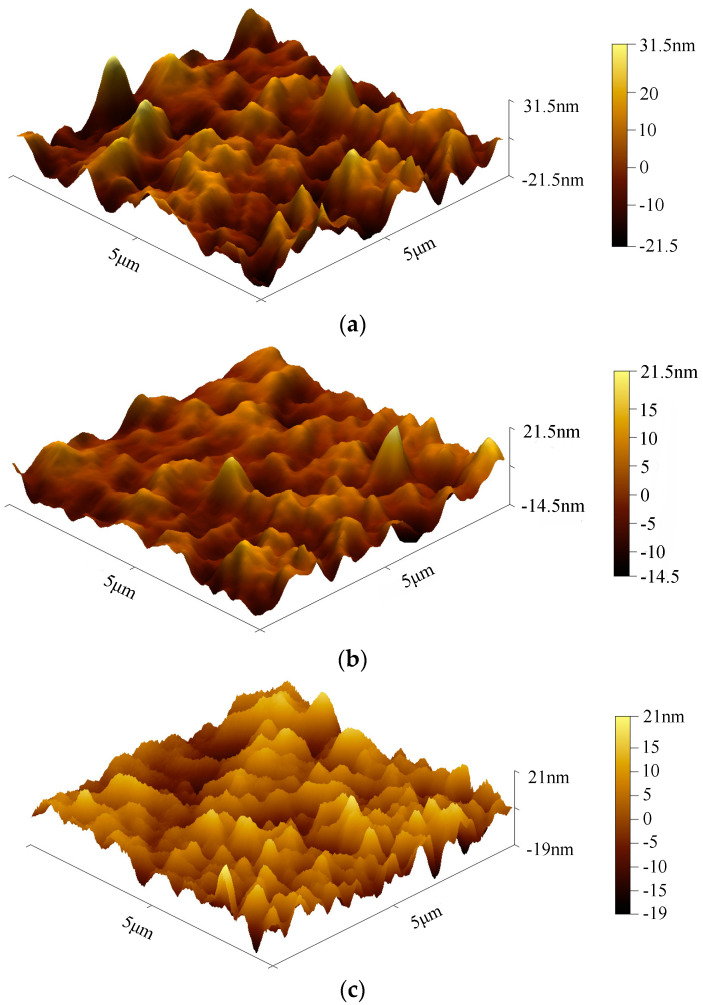
AFM image of surface topography of NiCr film. (**a**) AFM image of NiCr thin-film without surface temperature. (**b**) AFM image of NiCr thin-film with surface temperature. (**c**) AFM image of NiCr film after ion beam etching.

**Table 1 micromachines-13-00892-t001:** Resistance of thin film with different grid lengths prepared by PMS and PIBE.

Resistance Value/Ω	3 mm		4 mm		5 mm		6 mm	
	PMS	PIBE	PMS	PIBE	PMS	PIBE	PMS	PIBE
Total resistance	5127	3692	5314	3728	5449	3812	5829	3841
Single grid 1	183.5	128.3	268.1	178.2	306.1	220.4	410.2	291.3
Single grid 2	189.2	139.2	253.3	169.5	319.4	236.1	428.2	281.7
Single grid 3	188.6	131.8	258.6	171.6	311.8	223.8	430.6	279.6
Single grid 4	181.4	119.4	259.4	195.6	315.4	229.5	415.7	270.4
Single grid 5	190.2	134.1	264.9	180.2	309.9	236.7	420.5	280.3
Single grid range	8.8	19.8	14.8	26.1	13.3	16.3	20.4	20.9
Average error multiple	3.523	2.165	3.743	2.255	3.546	2.354	4.104	2.401

**Table 2 micromachines-13-00892-t002:** Geometric dimensions of thin-film resistor grids with different widths.

Sample	Width of Longitudinal Grid	Resistance Value: 1200 Ω; Length of Longitudinal Grid; 6 mm; Thickness of Longitudinal Grid: 800 nm			
		Transverse Grid Area	Number of Longitudinal Grids	Single Grid Resistance (Ω)	Electrode Area
1	0.05 mm	0.3 mm × 0.3 mm	7	165	2 mm × 2 mm
2	0.1 mm	0.4 mm × 0.4 mm	14	82.5	
3	0.15 mm	0.5 mm × 0.5 mm	21	55	
4	0.2 mm	0.6 mm × 0.6 mm	29	41.25	

**Table 3 micromachines-13-00892-t003:** Resistance of thin-film with different grid widths prepared by PMS and PIBE.

Resistance Value/Ω	0.05 mm		0.1 mm		0.15 mm		0.2 mm	
	PMS	PIBE	PMS	PIBE	PMS	PIBE	PMS	PIBE
Total resistance	7134	5645	5618	3606	5059	4696	4693	4501
Single grid 1	1031.4	800.2	395.2	255.2	244.1	223.1	168.1	159.2
Single grid 2	1024.6	803.1	406.1	258.1	236.4	228.4	173.4	157.4
Single grid 3	1019.2	792.8	410.9	268.0	229.1	226.8	165.9	138.4
Single grid 4	1014.4	811.6	401.1	266.7	250.6	219.1	159.8	168.1
Single grid 5	1026.8	808.7	393.8	246.9	243.7	217.3	167.1	171.5
Single grid range	17	18.8	17.1	21.1	21.5	11.1	13.6	33.1
Average error multiple	5.202	3.868	3.866	2.139	3.378	3.053	3.045	2.853

**Table 4 micromachines-13-00892-t004:** Geometric dimensions of thin-film resistor grids with different thicknesses.

Sample	Thickness of Longitudinal Grid	Resistance Value: 1200 Ω; Length of Longitudinal Grid: 6 mm; Width of Longitudinal Grid: 0.1 mm;			
		Transverse Grid Area	Number of Longitudinal Grids	Single Grid Resistance (Ω)	Electrode Area
1	800 nm	0.4 mm × 0.4 mm	14	82.5	2 mm × 2 mm
2	900 nm		16	73.3	
3	1000 nm		18	66	
4	1100 nm		20	59	

**Table 5 micromachines-13-00892-t005:** Resistance of thin film with different grid thicknesses prepared by PMS and PIBE.

Resistance Value/Ω	800 nm		900 nm		1000 nm		1100 nm	
	PMS	PIBE	PMS	PIBE	PMS	PIBE	PMS	PIBE
Total resistance	5726	3928	5618	3847	5464	3714	5328	3682
Single grid 1	403.4	286.7	344.7	235.9	310.1	203.8	321.4	188.6
Single grid 2	419.2	291.8	359.1	238.4	308.6	209.1	322.9	186.7
Single grid 3	416.5	281.6	362.7	241.7	300.8	214.8	318.2	176.8
Single grid 4	398.7	280.6	348.2	240.5	311.5	206.6	314.9	192.4
Single grid 5	402.9	276.8	351.6	250.0	297.5	196.8	308.4	184.6
Single grid range	20.5	15	18	14.1	14	18	14.5	15.6
Average error multiple	3.947	2.436	3.819	2.292	3.632	2.125	4.376	2.149

**Table 6 micromachines-13-00892-t006:** Level table of orthogonal test factors for ion beam etching.

Level	X	Y	Z
	Incident Angle (°)	Argon Flow (m^3^/s)	Substrate Negative Bias (V)
1	20	1 × 10^−5^	350
2	45	1.25 × 10^−5^	450
3	70	1.5 × 10^−5^	550

**Table 7 micromachines-13-00892-t007:** Orthogonal experimental design and experimental results of ion beam etching.

NO.	Incident Angle (°)	Argon Flow (sccm)	Substrate Negative Bias (V)	Etching Rate *v/*(nm/min)		Roughness Ra/(nm)	Resistivity *ρ* (μΩ·m)
				NiCr	Photoresist AZ6140		
1	20	6	350	19.8	18.2	35	1.21
2	20	7.5	450	25.0	23.4	38	1.57
3	20	9	550	31.5	27.8	41	1.68
4	45	6	450	23.6	22.6	31	1.10
5	45	7.5	550	29.1	26.3	33	1.19
6	45	9	350	20.9	18.1	28	1.06
7	70	6	550	27.6	26.6	37	1.46
8	70	7.5	350	20.2	17.4	32	1.16
9	70	9	450	26.1	24.2	36	1.32

**Table 8 micromachines-13-00892-t008:** Result and error of grid resistance of thin-film resistors of different sizes at 1200 Ω.

Resistance *R* (kΩ)		PIBE				PMS			
		Sample 1	Sample 2	Sample 3	Multiple of Average Error	Sample 1	Sample 2	Sample 3	Multiple of Average Error
width *d* (mm)	0.05	5.645	5.798	5.499	3.706	7.633	7.849	7.134	5.282
	0.1	3.606	3.823	3.937	2.157	5.726	5.328	5.618	3.631
	0.15	4.696	4.821	4.810	2.979	4.898	5.274	5.059	3.231
	0.2	4.501	4.134	4.794	2.730	4.852	5.091	4.693	3.065

**Table 9 micromachines-13-00892-t009:** Range analysis of thin-film etching experiment results.

	Film Etching Rate *v/*(nm/min)			Film Roughness Ra/(nm)			Resistivity ρ (μΩ·m)		
	X	Y	Z	X	Y	Z	X	Y	Z
*K* _1_	25.4	23.7	20.3	38.0	34.3	31.7	1.49	1.26	1.14
*K* _2_	24.5	24.8	24.9	30.7	34.3	35.0	1.12	1.33	1.33
*K* _3_	24.6	26.2	29.4	35.0	35.0	37.0	1.31	1.31	1.44
*R_j_*	0.9	2.5	9.1	7.3	0.7	5.3	0.37	0.06	0.30
	Influence level Z > Y > X			Influence level X > Z > Y			Influence level X > Z > Y		

**Table 10 micromachines-13-00892-t010:** Ion beam etching regression model analysis of variance.

		Freedom	SS	MS	F	P
etching rate, *v*	regression	3	134.55	44.85	191.67	0.000
	Residual Error	5	1.17	0.234		
	total	8	135.72			
Roughness, *Ra*	regression	3	56.83	18.94	1.37	0.354
	Residual Error	5	69.39	13.88		
	total	8	126.22			
Resistivity, *ρ*	regression	3	0.19408	0.06469	1.74	0.275
	Residual Error	5	0.18634	0.03727		
	total	8	0.38042			

## Data Availability

The data used to support the findings of this study are available from the corresponding author upon request.

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
