# Peer review of "Study of the Pattern Preparation and Performance of the Resistance Grid of Thin-Film Strain Sensors"

_micromachines, 2022, doi:10.3390/mi13060892_

Round 1

Reviewer 1 Report

In this research, the authors analyzed and compared two film patterning methods of photolithography magnetron sputtering and photolithography ion beam etching. The manuscript is well written and the data are explained well. However, the reviewer could not find any novelty claim section in the whole manuscript. If possible, it is better to clarify. And, a few more minor comments are added below.

1.      Acronyms should be given in first use e.g. PMS or PIBE

2.      Figure 2 Caption is haphazard, and difficult to understand for the reviewer.

3.      Strain sensors can the authors characterize more for real application.

4.      The bibliography needs to change to the proper format.

5.      Can the authors provide any microscopic image of the fabricated sensor?

6.      The introduction should be improved with more references.

7.      The conclusion should reflect all important findings.

Reviewer 2 Report

The manuscript reports on two film-patterning methods, namely photolithography magnetron sputtering (PMS) and photolithography ion bean etching (PIBE) with different grid lengths to analyze the electrical resistance and resistance strain coefficient. Overall, the study and analysis performed are satisfactory. However, many important points have been escaped from the investigations and discussions. The manuscript can be considered for publication after addressing the following major concerns.   

Comments:

  1. As per the design of Figure 1, why the number of grids is even, somehow contradicting Table 1? Additionally, the reviewer could not see identical resistance values for every longitudinal grid (especially for 5 mm) if the resistances were adjusted by varying the number of grids. Why?
  2. The second paragraph of the Experimental section is ambiguous. I recommend the authors rewrite using proper tense for experimental works.
  3. Please revise Figure 2 with the proper designation of each fabrication stage and its index.
  4. Between the two approaches for patterning techniques, how did the authors ensure the uniformity of patterned Ni-Cr alloy? Is there any resistance dependency with surface roughness?
  5. Please ensure the consistency between the figure and its legend in Figure 7b. Check others as well.
  6. The reviewer could not find any comparison of resistance strain coefficient under longitudinal and lateral strains. I strongly recommend including those investigations.
  7. The manuscript contains a lot of typographical errors and issues with standard units (e.g., units of resistance-line 73, page 2; pressure-line 88, page4; line 141-page5; and many more!). I recommend the authors to check such errors and issues throughout the manuscript and figure as well (especially, figure legends).

Reviewer 3 Report

This manuscript presented the application of photolithography magnetron sputtering and photolithography ion beam etching in the fabrication of resistance grids, and systematically studied various factors in the experiments. However, the manuscript is not of high quality in the presentation and essential analysis. Therefore, it is not suitable for publication in this journal at the current state. Some advice is as below.

(1)   Figure 2 does not match the corresponding text.

(2)   Why did you select the etching parameters in the table 2? Is there any theoretical basis for it?

(3)   The resistance value theoretically increases with the length, but decreases at 5 mm in Table 3. How to explain this?

(4)   How to ensure that the thickness of the two methods was the same in table 7, and have you measured the thickness?

(5)   Will photolithography magnetron sputtering and photolithography ion beam etching introduce other conductive impurities to change the resistance value?

(6)   The effect of process parameters on etching rate should be deeper analyzed.

Reviewer 4 Report

In this manuscript of micromachines-1738481, the effects of the geometric size of the grid obtained by photolithography magnetron sputtering and photolithography ion beam etching on the electrical performance of resistance value and resistance strain coefficient of the resistance grid are compared. And the effect of photolithography etching process on the surface quality of the film is discussed, and the etching process parameters are optimized.
These reseasches are valuable for fabricating resistance grid strain sensor of thin film, but

there are too many errors in the manuscrit. However, I am still willing to offer following suggestions to improve the manuscript.

1. To discuss the pattren transfering precision of the PMS and PIBE, and the lateral roughness of the grid of strain sensors should be measured.
2. Provide information of the material for insulating layer.
3. In the section of “3.4. Effect of ion beam etching on surface quality of Ni-Cr films” , the fig 12a and fi12b are out of order. To give the RMS roughness of the surface.
4. To offer the thickness of photoresist layer and the detailed process of stripping photoresist.
5. To discuss the influrence for annealing the Ni80Cr20 layer.
6. To explain why the PIBE method is better than the PMS method.
7. Extensive editing of English language and style required, especially in the imcomplete reference list.

Round 2

Reviewer 1 Report

I don't have any more comments for the authors. It can be accepted in its present form.

Reviewer 2 Report

The authors have revised the manuscript well. The manuscript can be considered for publication after making a few corrections.

1. Check the script for “strain” in figure 6b. The reviewer is wondering why the slope of the curve came out differently in the revised version with the same data!

2. Correct the unit of pressure as Pa (kPa or MPa or GPa, wherever necessary) and also pay attention to putting a “space” in-between numeric value and their units. This has to be corrected throughout the manuscript.  

Reviewer 4 Report

The authors have revised and improved the paper as the reviewer's suggestions, so I am willing to suggest the paper to be accepted in its current form.